# Establishment and Characterization of OFT and OFO Cell Lines from Olive Flounder (*Paralichthys olivaceus*) for Use as Feeder Cells

**DOI:** 10.3390/biology14030229

**Published:** 2025-02-24

**Authors:** Ja Young Jo, Ju-Won Kim, Eun Soo Noh, Yong-Ok Kim, Seung Pyo Gong, Hee Jeong Kong, Jae Hoon Choi

**Affiliations:** 1Biotechnology Research Division, National Institute of Fisheries Science, Busan 46083, Republic of Korea; jjy6556@naver.com (J.Y.J.); ogamzar@korea.kr (J.-W.K.); laperm@korea.kr (E.S.N.); yobest12@korea.kr (Y.-O.K.); heejkong@korea.kr (H.J.K.); 2Major in Aquaculture and Applied Life Science, Division of Fisheries Life Science, Pukyong National University, Busan 48513, Republic of Korea; gongsp@pknu.ac.kr

**Keywords:** testicular cell, ovarian cell, germline stem cell, co-culture

## Abstract

Germline stem cells (GSCs) have applications in aquaculture. As a preliminary step toward the development of an in vitro GSC culture system for olive flounder (*Paralichthys olivaceus*), testicular and ovarian cell lines (OFT and OFO, respectively) of this species were established and characterized. Coculture of OFT or OFO with enriched *P. olivaceus* male GSCs showed that these cell lines were capable of maintaining male GSCs as feeder cells. Our research will contribute to the development of an in vitro GSC system for *P. olivaceus*.

## 1. Introduction

Germline stem cells (GSCs) transmit their genetic information to the next generation with the exchange of complex signals with somatic cells [1,2]. Using this remarkable trait, surrogate broodstock technology, which is the production of donor-derived gametes in surrogates by injection of donor-derived GSCs in recipients, can be used in aquaculture for the production of offspring carrying superior genetic information, long-term storage of invaluable fish, and the improvement of productivity by reducing the time required for production of gametes [3]. However, there is a crucial difficulty that the number of GSCs harvested from donor individuals is not sufficient for injection into recipients [4]. In vitro GSC culture systems represent a viable alternative to overcome this obstacle.

GSCs have been cultured from numerous teleost fish species. The first male GSC line, SG3, was established from medaka (*Oryzias latipes* [5]). The SG3 line was cultured for over 2 years without feeder cells and maintained expression of germ cell-specific genes, including *vasa*, *dazl*, *piwi*, and *c-kit*. There have been reports on the in vitro culture of male GSCs from teleost fish species without feeder cells, including swamp eel (*Monopterus albus* [6]), Chinese hook snout carp (*Opsariichthys bidens* [7]), orange-spotted grouper (*Epinephelus coioides* [8]), and tiger puffer fish (*Takifugu rubripes* [9]). However, there have been no reports of the successful production of fertile sperm or offspring from cultured GSC lines without feeder cells. On the other hand, in zebrafish (*Danio rerio*), testicular cells containing male GSCs and testicular somatic cells were cultured for 1.5 months and successfully produced offspring following injection of grafted aggregates into recipients [10]. Not only male GSCs, but also female *D. rerio* GSCs were cultured with a feeder cell line for 3 weeks, and offspring were produced by surrogate broodstock technology [11]. GSCs from male rainbow trout (*Oncorhynchus mykiss*) were cultured with the Sertoli cell line for 28 days and successfully produced functional eggs and sperm by surrogate broodstock technology [4].

Olive flounder (*Paralichthys olivaceus*), as a major aquaculture species in Japan, China, and the Republic of Korea (hereafter, Korea), is a commercially important fish for the aquaculture industry [12]. Indeed, there have been a number of trials in Korea to improve its productivity through selective breeding [13], triploid induction [14,15], and CRISPR/Cas9-mediated myostatin disruption [16]. However, there have been no studies regarding in vitro GSC culture in *P. olivaceus*.

As a preliminary study to develop a stable in vitro *P. olivaceus* GSC culture system, we established olive flounder testicular and ovarian cell lines (OFT and OFO, respectively). Then, we characterized the two cell lines and evaluated them as feeder cells via coculture with male *P. olivaceus* GSCs enriched by Percoll density gradient centrifugation. The two established cell lines are expected to be used importantly in the development of in vitro culture systems for *P. olivaceus* GSCs through further studies.

## 2. Materials and Methods

### 2.1. Fish and Ethics Statement

Olive flounders (*P. olivaceus*) were raised at the National Institute of Fishery Sciences (NIFS) in Busan, Korea. They were sustained in 1-ton flow through tanks adjusted to 19 ± 0.3 °C under a natural photoperiod with an electronic timer and fluorescent lighting. They were fed twice a day with commercial extruded pellets (Daebong LS Co., Ltd., Jeju, Korea). Animal experiments were performed in accordance with the Animal Protection Act of the Ministry of Agriculture, Food, and Rural affairs, Republic of Korea, and were approved by the Institutional Animal Care and Use Committee of the NIFS (2024-NIFS-IACUC-47).

### 2.2. Primary Culture and Subculture

To establish OFT and OFO cell lines, 8-to-10-month-old *P. olivaceus* males and females were anesthetized in seawater containing 0.1% (*v*/*v*) 2-phenoxy ethanol (Sigma-Aldrich, St. Louis, MO, USA) for 10 min, after which they were exposed to 70% ethanol (Supelco, Burlington, MA, USA) to prevent microbial contamination. Then, the testes and ovaries were extracted and washed three times with Dulbecco’s phosphate-buffered saline (DPBS; Gibco, Grand Island, NY, USA). Samples of 0.5 g testes and ovaries were placed into 100 mm Petri dishes (Corning, Corning, NY, USA) and dissected with a surgical blade (No. 10; Paragon, Sheffield, UK). Subsequently, these tissue samples were digested with 10 mL 0.25% (*v*/*v*) trypsin-EDTA (Gibco) for 2 h with pipetting every 10 min. Digested cells were filtered sequentially through 100 µm and 40 µm cell strainers (Falcon, Durham, NC, USA). To inactivate trypsin-EDTA, the same volume of modified ESM2 medium (mESM2) without bFGF, serum, and embryo extract (EE) was added and cells were harvested by centrifugation (400× *g*, 5 min, 20 °C). Finally, aliquots of 4.5 × 10^5^ cells were seeded into 6-well plates (Corning), which were precoated with 0.2% gelatin (Sigma-Aldrich) and cultured in an incubator adjusted to 20 °C. The culture medium was changed every 3 days. When the cells reached about 90% confluence, they were sub-cultured into new gelatin-coated 25 cm^2^ culture flasks at a 1:2 ratio.

### 2.3. Cell Culture Medium

To culture OFT and OFO, mESM2 without embryo extract (EE) was used. However, enriched male GSCs were cocultured with OFT or OFO in complete mESM2. The composition of mESM2 was almost the same as ESM2, originally designed for fish embryonic stem cell culture, except for the use and concentration of *P. olivaceus* EE and *P. olivaceus* serum. Briefly, mESM2 was composed of Dulbecco’s Modified Eagle’s Medium (DMEM; Gibco, Grand Island, NY, USA) supplemented with 15% (*v*/*v*) fetal bovine serum (Gibco), 20 mM hydroxyethyl piperazine ethane sulfonic acid (Sigma-Aldrich, St. Louis, MO, USA), 1 mM nonessential amino acids (Gibco), 1 mM sodium pyruvate (Gibco), 2 nM sodium selenite (Sigma-Aldrich), 100 μM 2-mercaptoethanol (Gibco), 10 ng/mL human basic fibroblast growth factor (bFGF; Gibco), 1% (*v*/*v*) *P. olivaceus* serum, 50 μg/mL EE, and 1% (*v*/*v*) P/S. Serum and EE from *P. olivaceus* were prepared with reference to previous studies [17,18].

### 2.4. Validation of Cell Type from OFT and OFO

To validate cell types from OFT and OFO, we investigated the expression of germ cell-specific genes (*vasa*, *nanos2*, *scp3*) and gonadal somatic cell-specific genes (*wt1*, *gsdf*, *fgf2*). Total RNA was extracted from OFT and OFO at passage 30 using an RNeasy Micro kit (Qiagen, Hilden, Germany) according to the manufacturer’s instructions. Subsequently, 100 ng total RNA was used for cDNA synthesis using SuperScript™ IV VILOTM Master mix with ezDNase (Invitrogen, Vilnius, Lithuania) according to the manufacturer’s instructions. PCR was performed using Accupower PCR Master Mix (Bioneer, Daejeon, Republic of Korea). The primer sequences used in this study are presented in Table 1. RT-PCR conditions consisted of an initial denaturation step for 5 min at 95 °C followed by 35 cycles of 95 °C for 30 s, 60 °C for 30 s, 72 °C for 30 s, and a final extension step at 72 °C for 5 min.

### 2.5. Validation of Cell Origin from OFT and OFO

Mitochondrial DNAs were extracted from OFT, OFO, and the fins of *P. olivaceus* using a QIAamp micro kit (Qiagen) according to the manufacturer’s instructions. These samples were analyzed using conventional PCR with 2× premixture (SolGent, Daejeon, Republic of Korea) and cytochrome c oxidase subunit 1 (*COI*) or *16s rRNA* primers (Table 1). PCR conditions consisted of an initial denaturation step for 10 min at 95 °C followed by 35 cycles of 95 °C for 50 s, 50 °C for 50 s, 72 °C for 50 s, and a final elongation step for 7 min at 72 °C. Then, the PCR products were detected by electrophoresis in 1.5% (*w*/*v*) agarose gels (Biosesang, Yongin, Republic of Korea) and visualized using a gel documentation system (Uvitec, Cambridge, UK). Subsequently, the amplified PCR products were analyzed by automated DNA sequencing (ABI 3730XL DNA analyzer; Applied Biosystems, Foster City, CA, USA) and compared to sequences registered in GenBank using the NCBI Basic Local Alignment Search Tool (BLAST).

### 2.6. Karyotyping

To confirm the chromosome complements of the two established cell lines, OFT and OFO were cultured and subjected to karyotype analysis. OFT and OFO at passage 30 were inoculated into 25 cm^2^ culture flasks and incubated for 24 h at 20 °C. Cells were treated with 3 μg/mL colchicine (Sigma-Aldrich) for 150 min at 20 °C, and then the culture medium was removed and cells were washed with 1 mL DPBS (Gibco). Cultured cells were detached by adding 2 mL 0.25% trypsin-EDTA (Gibco). Detached cells were transferred into 15 mL conical tubes and 2 mL cell culture medium was added for inactivation of trypsin-EDTA. The cells were harvested by centrifugation (400× *g*, 5 min, RT), gently resuspended in 3 mL 0.075 M KCl (Shinyo Pure Chemical, Osaka, Japan), and incubated for 20 min at room temperature. Subsequently, the cells were harvested via centrifugation under the same conditions and resuspended in 3 mL Carnoy solution consisting of 3:1 methanol:acetic acid (Daejung, Seoul, Republic of Korea) at room temperature for 1 h with fresh solution exchange every 30 min. The cells were harvested by centrifugation (400× *g*, 5 min, RT) and resuspended in 200 µL Carnoy solution. Aliquots of 20 µL cell suspension were dropped onto slide glasses, which were then immersed in 100% (*v*/*v*) acetic acid (Daejung) for 1 day, blotted with tissues, and stained for 8 min with Giemsa staining solution consisting of 8% (*v*/*v*) Giemsa stain (Gibco) in Gurr’s buffer (Gibco). After washing with distilled water, the slide glasses were dried completely at room temperature. Finally, chromosomes of metaphase spreads were observed under a microscope (Zeiss, Jena, Germany) with oil immersion optics (Merck, Darmstadt, Germany) at ×1000 magnification. To determine the percentage karyotype normality for OFT and OFO, 63 and 70 photographs of each karyotype were taken, respectively, and the normality percentages were calculated as follows: (number of metaphase spreads/number of cells and metaphase spreads) × 100 [19].

### 2.7. Sex Analysis

Genomic DNA was isolated from OFT and OFO using a QIAamp micro kit for genomic DNA extraction (Qiagen). These samples were analyzed using a QuantStudio 5 Real-Time PCR System (Applied Biosystems) with the following steps: 95 °C for 5 min and 35 cycles of 95 °C for 15 s and 60 °C for 1 min. SNP genotyping analysis was carried out using 2× Real-Time PCR Master Mix (Attoplex, Gyeonggi-do, Republic of Korea) in a total volume of 20 μL, with each primer at a final concentration of 0.5 μM, while the probe primer was used at a final concentration of 0.25 μM. The SNP primer markers used are shown in Table 2. Positive controls were prepared by extracting genomic DNA from the fins of male and female *P. olivaceus*, which were managed by NIFS.

### 2.8. Electroporation

Aliquots of 1 × 10^6^ OFT or OFO cells were collected by centrifugation at 400× *g* for 5 min. Subsequently, they were resuspended in 100 µL Opti-MEM (Gibco) containing 10 µg green fluorescent protein (GFP) expression vector pEGFP-c1. Cells were transferred into cuvettes (NEPA Electroporation Cuvettes, 2 mm gap; Nepa Gene Co. Ltd., Chiba, Japan) and electroporated at 160 V using a NEPA21 Super Electroporator (NEPA GENE Co. Std., Chiba, Japan). Immediately after electroporation, 1 mL mESM2 was added for stabilization and the cells were transferred into 6-well plates (Corning). These cells were cultured in an incubator at 20 °C. At 48 h after electroporation, green fluorescent signals were examined using a fluorescence microscope (Axio Vert A1; Zeiss). Transfection efficiency was quantified using ImageJ (ver. 1.51k; National Institutes of Health, Bethesda, MD, USA).

### 2.9. Enrichment of Male P. olivaceus GSCs by Percoll Density Gradient Centrifugation

Quantitative real-time PCR (qRT-PCR) was performed to determine optimal Percoll centrifugation layers for isolating male *P. olivaceus* GSCs. First, singly digested total testicular cells were resuspended in 300 µL mESM2 and loaded onto a discontinuous 5-layer gradient of Percoll solution consisting of 1 mL each of 20%, 30%, 40%, 50%, and 60% in 15 mL conical tubes (SPL) and centrifuged at 800× *g* for 30 min. The cells harvested from each density fraction were washed with DPBS. Total RNA extraction from cells isolated from each Percoll centrifugation layer and cDNA synthesis were performed using a RNeasy Micro kit (Qiagen) and SuperScript™ IV VILOTM Master mix with ezDNase (Invitrogen) according to the method outlined above. Then, 100 ng cDNA was subjected to qRT-PCR with Fast SYBR Green Master Mix (Applied Biosystems) to evaluate the level of *plzf* expression from each of Percoll centrifugation layer. Amplification and detection were conducted using QuantStudio 3 (Applied Biosystems) with the following steps: 50 °C for 2 min, 95 °C for 10 min, and 40 cycles of 95 °C for 15 s and 60 °C for 30 s. Relative quantification of qRT-PCR data was carried out using the 2^−ΔΔCt^ method, where Ct = threshold cycle for target amplification, ΔCt = Ct target gene (*plzf*) − Ct internal reference (*18s rRNA*), and ΔΔCt = ΔCt sample − ΔCt calibrator [20].

### 2.10. Coculture with Germ Cells

To evaluate their suitability as feeder cells, OFT and OFO were cocultured with male *P. olivaceus* GSCs. First, 1 × 10^5^ OFT or OFO cells were seeded on 0.2% gelatin-coated 24-well plates (Corning) and cultured overnight. Subsequently, they were mitotically inactivated by treatment with 10 μg/mL mitomycin C (Roche, Mannheim, Germany) for 3 h. Mitotically inactivated cells were washed twice with DPBS (Gibco). Then, 1 × 10^5^ male *P. olivaceus* GSCs isolated via Percoll density gradient centrifugation were seeded into 24-well plates, in which mitotically inactivated OFT or OFO had been cultured. Finally, they were cocultured for 7 days in an incubator adjusted to 20 °C and culture medium was exchanged every 3 days. The experiment was conducted in 3 repetitions.

### 2.11. Statistics

Data are presented as the mean ± standard deviation (SD) (*n* = 3). Data were analyzed via one-way analysis of variance (ANOVA), followed by a Duncan’s multiple range test using SPSS (version 19.0; SPSS Inc., Chicago, IL, USA). In all analyses, *p* < 0.05 was taken to indicate statistical significance.

## 3. Results

### 3.1. Primary Cell Culture and Gene Expression Analysis of OFT and OFO

OFT and OFO were established from testes and ovaries of *P. olivaceus* through primary cell culture and subculture. At passage 5, fibroblast-like and epithelial-like cells were dominant in OFT and OFO (Figure 1A,B). By contrast, cobblestone-like and fibroblast-like cells were dominant and other epithelial-like cells were less abundant (Figure 1C,D). The expressions of various genes, including germ cell markers (*vasa*, *nanos2*, and *scp3*) and somatic cell markers (*wt1*, *gsdf*, and *fgf2*), were analyzed in OFT and OFO cells at passage 30. Obvious *wt1* and *fgf2* expression was observed in OFT, while weak *wt1* expression and obvious *fgf2* expression were observed in OFO (Figure 1E). While *vasa*, *nanos2*, *scp3*, and *gsdf* gene expression was observed in testicular and ovarian tissues, they were not detected in OFT and OFO.

### 3.2. Species Identification from OFT and OFO

To identify the species of OFT and OFO, the cytochrome c oxidase subunit 1 (*COI*) and *16s rRNA* region were amplified via PCR. As shown in Figure 2, *COI* amplification products of OFT and OFO were confirmed at 650 bp, which was the same size as in the positive control. Similarly, *16s rRNA* amplification products of OFT and OFO were confirmed at 542 bp, and they showed the same size as in the positive control. In addition, the sequences of *COI* and *16s rRNA* showed matches of 100% with the corresponding *P. olivaceus* sequences in GenBank (Appendix A).

### 3.3. Karyotype Analysis of OFT and OFO

Representative images of chromosome sets (2*n* = 48) from OFT and OFO are shown in Figure 3. The numbers of OFT and OFO chromosomes showed a wide distribution between more than 40 and less than 50, with a modal peak at 48 chromosomes for each cell type. The percentage karyotype normality for OFT and OFO was 48% and 41.3%, respectively.

### 3.4. Sex Identification Analysis in OFT and OFO Cells

Due to environmental factors, *P. olivaceus* can undergo sex reversal into phenotypic males. For this reason, to determine that OFT and OFO truly originated from maled and femaled, SNP markers (5227A/T, 8483C/T, 5512C/T), which were developed by NIFS, were used. The markers 5227A/T, 8483C/T, and 5512C/T were detected from male *P. olivaceus* (Figure 4). Similarly, the SNP results from OFT were similar to male *P. olivaceus*. Only the 5512 SNP marker was detected in both female *P. olivaceus* and OFO. These results demonstrate that OFT originated from male *P. olivaceus* and OFO originated from female *P. olivaceus*.

### 3.5. Transfection of Green Fluorescent Protein Vector into OFT and OFO Cells

Transfection was generally proceeded by transfection reagent or electroporation. In this study, we chose the electroporation method. To determine transfection efficiency and gene expression, OFT and OFO cells were transfected with pEGFP-c1 via electroporation. Both cell types transfected with pEGFP-c1 exhibited strong green fluorescent signals at 48 h after transfection (Figure 5). The transfection efficiencies of OFT and OFO cells were approximately 35.63% and 34.64%, respectively. Therefore, OFT and OFO cells are valuable for exogenous gene expression, which is crucial for both fundamental research and biotechnological applications.

### 3.6. Short-Term Coculture with Enriched Male P. olivaceus GSCs and OFT or OFO

To determine the optimal conditions for male GSC enrichment, total testicular (TT) cells were separated via Percoll density gradient centrifugation. Subsequently, *plzf* expression levels in the cells separated from the density fractions were evaluated. qRT-PCR analysis revealed that *plzf* expression levels were significantly higher in the top 20% and 20–30% density fractions (Figure 6A). In addition, these density fractions showed significantly higher *plzf* expression levels than TT. Therefore, the top 20% and 20–30% density fractions were selected for enrichment of male *P. olivaceus* GSCs. Finally, short-term coculture was performed to evaluate OFT and OFO as feeder cells. The experiment is shown schematically in Figure 6B. GSCs were determined based on morphological features, including their round or oval-shaped flattened nucleus [21]. On day 7, five or seven male *P. olivaceus* GSCs attached to two cell lines, which was not observed in cultures of OFT or OFO alone (Figure 6C). Those results indicated that male *P. olivaceus* GSCs can attach to the two established cell lines.

## 4. Discussion

In early passages, OFT and OFO were composed of fibroblast-like and epithelial-like cells, while in late passages, they were composed of cobblestone-like, fibroblast-like, and epithelial-like cells. These results were similar to a previous study in which *P. olivaceus* Sertoli cell lines in early passages showed fibroblast- and epithelial-like cells [22]. In gonadal cell lines, fibroblast- and epithelial-like cells were assumed to act as feeder cells [10]. Indeed, our RT-PCR results revealed that both cell lines expressed gonadal somatic cell-specific genes, including *wt1* and *fgf2*. Among these, *wt1* is a Sertoli cell marker [23]. Therefore, the OFT cell line was estimated to have been derived from Sertoli cells. Although *wt1* is a representative Sertoli cell marker, OFO showed relatively weak expression of this gene. Theca cells were reported to express *wt1* [24]. In addition, *wt1* expression was observed in a half-smooth tongue sole (*Cynoglossus semilaevis*) ovarian cell line [25]. Interestingly, both cell lines expressed *fgf2*, which induced self-renewal of male and female GSCs [11,26]. In addition, *fgf2* is expressed in Sertoli cells [27] and granulosa cells [28]. Taken together, these observations suggest that OFT cells originated from Sertoli cells and OFO cells originated from theca cells and granulosa cells.

Amplification and sequencing of *COI* and *16S rRNA* have been used to investigate the origin of cell lines [22,29,30]. Amplified PCR products from OFT and OFO mitochondrial DNA corresponded in size to *P. olivaceus* mitochondrial DNA. In addition, we confirmed that *COI* and *16S rRNA* genes derived from OFT and OFO were 100% identical to those of *P. olivaceus*. Therefore, we confirmed that OFT and OFO were truly derived from *P. olivaceus*.

Karyotype analysis is an indicator of the unique state of cells [31]. The normal *P. olivaceus* chromosome number is 2*n* = 48 [32,33]. In the present study, OFT showed 47.6% normality and OFO showed 41.3% normality, indicating that about half of the cultured cells maintained the parental karyotype. However, among the abnormal chromosome number, similar numbers of chromosomes such as 48 ± 3 may have been the result of the addition or removal of chromosomes from nearby cells [34]. In addition, an established *P. olivaceus* Sertoli cell line (POSC) showed 48% normality in a previous study [22]; our results were similar. Therefore, the karyotype results demonstrated the stable maintenance of OFT and OFO in culture.

Under special environmental conditions, female teleost fish including *P. olivaceus* can undergo sex reversal into phenotypic males, so-called pseudo-males [6,35]. As pseudo-male gonad-derived cell lines express Sertoli cell markers [25], SNP markers developed by NIFS were used to identify genetic sex. These SNP markers were previously used to identify the genetic sex of *P. olivaceus* [36]. The results of our SNP analysis revealed that OFT and OFO were derived from XY male testis and XX female ovary, respectively.

Transfection of exogenous DNA into established cell lines is useful for both basic research and biotechnological applications. Generally, chemical transfection reagents have been used for transfection of DNA into cultured cell lines. Three sturgeon heart, gonad, and head kidney cell lines were transfected with pEGFP-N1 using various commercial transfection reagents [37]. However, three cell lines exhibited transfection efficiency of less than 5%. SIMH derived from the heart of milkfish (*Chanos chanos*) and SIGE derived from the eye of grouper (*Epinephelus coioides*) were transfected with pEGFP-N1 using LipofectAMINE™ 2000 [38]. However, both cell lines exhibited transfection efficiencies of about 0.0125%. POSCs were transfected with pEGFP-N3 using LipofectAMINE™ 2000 and exhibited 10% transfection efficiency [22]. In this study, however, the pEGFP-C1 GFP expression vector was transfected into OFT and OFO by electroporation and showed relatively high transfection efficiency. Although exogenous DNA size and concentration were different, our results demonstrate that electroporation was more efficient for transfection in OFT and OFO than chemical agents.

In mammals, male GSCs are generally isolated by fluorescence-activated single cell sorting (FACS) and magnetic-activated cell separation (MACS). However, the use of FACS or MACS would require preparation of male GSC surface antigen-specific antibodies. To the best of our knowledge, there have been no reports of male *P. olivaceus* GSC surface antigen-specific antibodies, so we used Percoll density gradient centrifugation. Previously, male GSCs were isolated via Percoll density gradient centrifugation from various teleost fish species, including marine four-eyed sleeper (*Bostrychus sinensis* [39]), blue catfish (*Ictalurus furcatus* [40]), curimbatá (*Prochilodus lineatus* [41]), and Jundiá (*Rhamdia quelen* [42]). In addition, male GSCs of *P. olivaceus* were isolated using this method in a previous study, and the authors determined that the optimal layers were the top 20% and 20–35% layers based on cell morphology [43]. The *plzf* gene is a well-known GSC marker that plays a role in maintaining the undifferentiated state of GSCs [44]. Our observations are consistent with this, with the optimal layers being the top 20% and 20–30%, as determined by the *plzf* expression level of each of density fraction. Taken together, we conclude that the top 20% and 20–30% layers are the optimal layers for enrichment of male *P. olivaceus* GSCs.

The proliferation and survival of male GSCs are regulated by gonadal somatic cells [4]. Therefore, enriched GSCs were cocultured for 7 days with OFT or OFO to evaluate their utility as feeder cells. On day 7, cocultured GSCs were found to have attached to OFT or OFO. In the previous study, adhesion between the GSCs and gonadal somatic cells have revealed that it is critical for maintenance GSCs [45,46]. However, with increasing time in culture, cocultured cells became detached from the plates. This was probably because the culture conditions were not optimized for long-term culture. In a previous study, male *D. rerio* GSCs proliferated in culture medium supplemented with GDNF, insulin growth factor-1 (IGF-1), and bFGF [10]. Therefore, further studies are required to evaluate the effects of other growth factors for a stable *P. olivaceus* GSC in vitro culture system.

## 5. Conclusions

In this study, we not only successfully established OFT and OFO from male and female *P. olivaceus*, but also established conditions for enrichment of *P. olivaceus* male GSCs by Percoll density gradient centrifugation. OFT demonstrated that it originated from Sertoli cells of XY male *P. olivaceus*. On the other hand, OFO demonstrated that it originated from granulosa and theca cells of XX female *P. olivaceus*. Finally, coculture results demonstrated that those two established cell lines have the ability to attach to male *P. olivaceus* GSCs.

## Figures and Tables

**Figure 1 biology-14-00229-f001:**
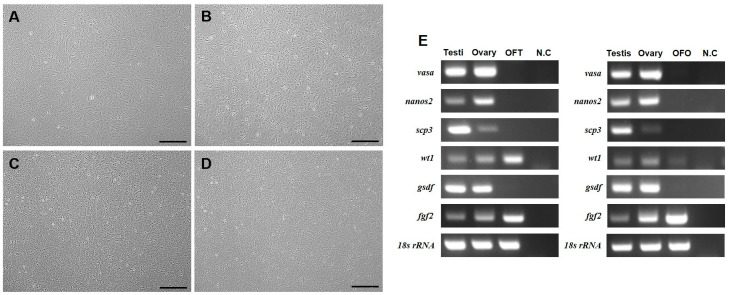
Morphology and characterization of olive flounder testicular cells (OFT) and olive flounder ovarian cells (OFO). (**A**) The representative image of OFT at passage 5. (**B**) The representative image of OFO at passage 5. (**C**) The representative image of OFT at passage 20. (**D**). The representative image of OFO at passage 20. (**E**) RT-PCR results from OFT and OFO at passage 30. Scale bar = 400 μm.

**Figure 2 biology-14-00229-f002:**
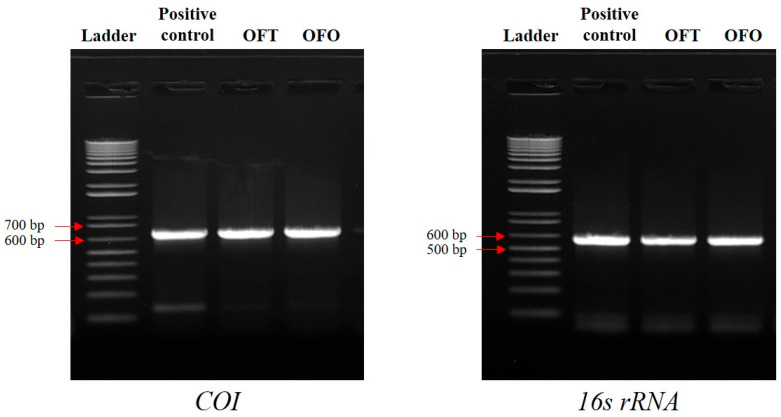
Species identification from OFT and OFO. Amplification of *COI* and *16S rRNA* regions from two cell lines. As positive controls, genomic DNA extracted from fin tissues of olive flounder was used. Arrows indicate ladder sizes.

**Figure 3 biology-14-00229-f003:**
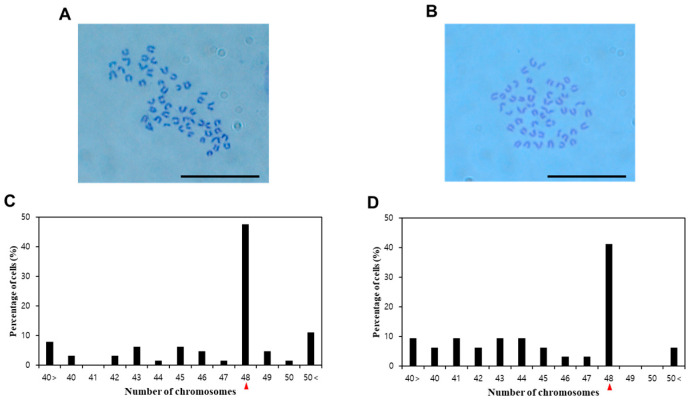
Karyotype analysis of OFT and OFO. Representative images of metaphase spreads from (**A**) OFT and (**B**) OFO. Normality test results (2*n* = 48) from (**C**) OFT and (**D**) OFO. Red arrowheads indicate the number of diploid *P. olivaceus* chromosomes. Scale bar = 20 μm.

**Figure 4 biology-14-00229-f004:**
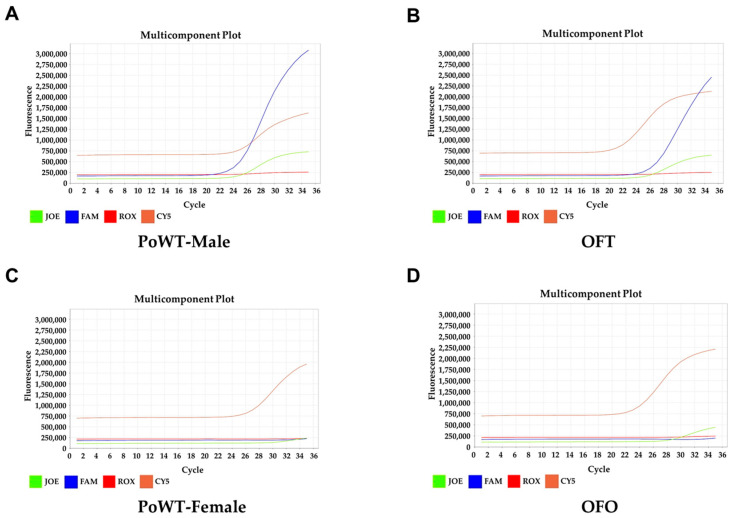
Sex identification analysis. The genetic sex of OFT and OFO was determined using SNP markers (5512T, 5227A, 8483C) by real-time PCR with validation by comparative analysis with wild-type *P. olivaceus*. (**A**) Male wild-type *P. olivaceus*. (**B**) OFT at passage 20. (**C**) Female wild-type *P. olivaceus*. (**D**) OFO at passage 20.

**Figure 5 biology-14-00229-f005:**
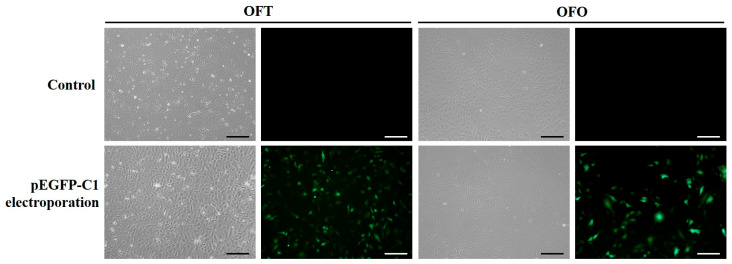
Transfection of green fluorescent protein expression vector into OFT and OFO cells. OFT and OFO were electroporated with the green fluorescent protein (GFP) expression vector pEGFP-C1. After electroporation, the two cell lines were cultured for 48 h and observed by fluorescent microscopy. Scale bar = 200 μm.

**Figure 6 biology-14-00229-f006:**
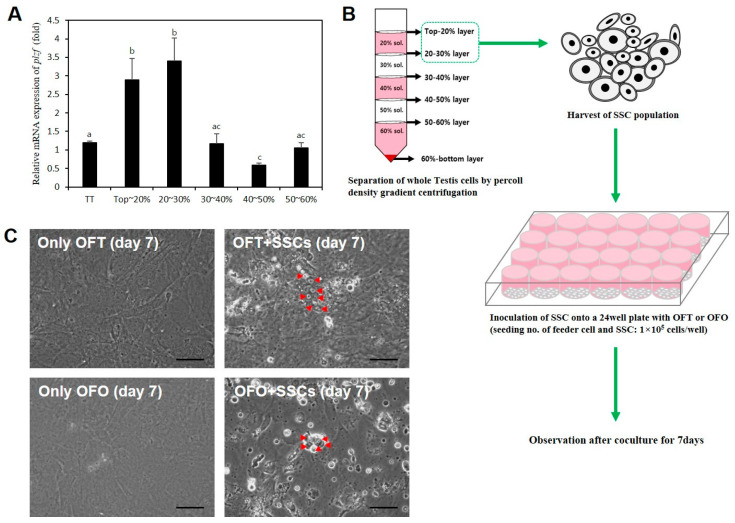
Evaluation of the enrichment of male GSCs via Percoll density gradient centrifugation. (**A**) Relative expression of *plzf* from each density fraction. ^abc^ Different letters indicate significant differences (*p* < 0.05). (**B**) Schematic representation of experimental procedures. Two different cell populations from the top–20% and 20–30% layers after Percoll density gradient centrifugation showed cell attachment after coculture for 7 days. (**C**) The representative images of coculture with GSCs or without. Scale bar = 50 μm.

**Table 1 biology-14-00229-t001:** Primer sequences used in this study.

Gene	Primer Sequence (5′>3′)		Product Size (bp)	Accession Number
*vasa*	Forward	CAGGACAGCACAGCGAAGAG	141	JQ070418.1
	Reverse	GCAACAAGCTAAACAGCAAATAAGAG		
*nanos2*	Forward	CGGACCACTGTCGCTTCTG	159	XM_020087405.1
	Reverse	ACCGGCGTGTGTGTGCTT		
*scp3*	Forward	TGGCTACCGTCCGCAAGT	154	XM_020090502.1
	Reverse	CGATATGAACACGAACCAAATTAAGT		
*wt1*	Forward	TGTTTGGTTGCCACAATCCTT	101	XM_020098636.1
	Reverse	CAGCTGAGATGCCATTTGGTATAC		
*gsdf*	Forward	CCTGAGATGAACACTGTGCAATG	151	KY123266.1
	Reverse	GCACGGAGGAAATGATGACTGT		
*fgf2*	Forward	AAGACAAAGAAGAAGATGTGAAGACAGA	200	XM_020105694.1
	Reverse	AAGGCACCTGGCTGCAGTT		
*18s rRNA*	Forward	ATGGCCGTTCTTAGTTGGTG	218	EF126037.1
	Reverse	CACACGCTGATCCAGTCAGT		
*plzf*	Forward	TCCTCTTCCACCGCAACAG	79	XM_020097052.1
	Reverse	GCATACTCCAAAATCTGCTGAAAA		
*COI*	Forward	TGTAAAACGACGGCCAGTCAACCAACCACAAAGACATTGGCAC	650	NC_082846.1
	Reverse	CAGGAAACAGCTATGACACTTCAGGGTGACCGAAGAATCAGAA		
*16s rRNA*	Forward	CCGGTCTGAACTCAGATCACGT	542	NC_082846.1
	Reverse	CGCCTGTTTATCAAAAACAT		

**Table 2 biology-14-00229-t002:** Primer sequences for SNP analysis to identify genetic sex.

Primer Name	Primer Sequence (5′>3′)	Product Size (bp)
5227A	TCAAATGGCATAGATGGACA	116
5227T	TCAAATGGCATAGATGGACT	
5227	TCATGCGGTAATTGCTTTGTA	
5227-FAM	FAM-ATGCGGACGAGTAGTTCATTCGAA-EBQ	170
8483C	AGTCCACATTTACGAGAGTTC	
8483T	AGTCCACATTTACGAGAGTTT	
8483	GATGAACGAGAAATTAGATTTCCCCG	
8483-JOE	JOE-GCTTACGGAGGGCAGCTAGCAACGA-EBQ	
5512	ATGGCCTGGATTGCATCAAC	154
5512T	ATGGCCTGGATTGCATCAAT	
5512	CTTAGCATAAGGAACCCACGTC	
5512-CY5	CY5-TTCCGACGCGTTGTACACGGGCAA-EBQ	

## Data Availability

The datasets used or analyzed during the current study are available from the corresponding author upon reasonable request.

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
