# Peer review of "Establishment and Characterization of OFT and OFO Cell Lines from Olive Flounder (Paralichthys olivaceus) for Use as Feeder Cells"

_biology, 2025, doi:10.3390/biology14030229_

Round 1

Reviewer 1 Report

Comments and Suggestions for Authors

In this paper, the author established the testicular and ovarian cell lines of Paralichthys olivaceus, which were named OFT and OFO respectively. The detection of these two cells by RT-PCR showed that the cells only expressed several gonadal somatic cell markers (wt1, fgf2) and did not express germ cell markers (vasa, nanos2, scp3). Through COâ… , 16s rRNA gene sequencing, the homologous match with NCBI sequence was up to 100%. Meanwhile, SNP analysis showed that OFT came from XY male P. olivaceus and OFO came from XX female P. olivaceus. The types of OFT cells were identified as Sertoli cells and OFO cells as granulosa cells and theca cells. Coculture of OFT or OFO with GSCs separated by Percoll was amid to verify the capable as feeder cells.

Minor editorial revisions will be needed.

  1. Line54-56: “However, there have been no 54 reports of the successful production of fertile sperm or offspring from cultured GSC without feeder cells.” Please check this article (Establishment of a Southern catfish (Silurus meridionalis) spermatogonial stem cell line capable of sperm production in vitro) to read if they fit your sentence. Meanwhile, this relates to another function of feeder cells, inducing stem cell differentiation. Should you cite examples of feeder cells maintaining stem cells better in vitro?
  2. The line 233 describes passage 5, and the algebra labeled in the figure note of figure1 B is passage 20.
  3. Line 243: “Morphology and characterization of olive flounder testicular cells (OFT) and olive flounder 243 ovarian cells (OFO).” should be bolded, like this “Morphology and characterization of olive flounder testicular cells (OFT) and olive flounder 243 ovarian cells (OFO).
  4. Line 250: “Figure 2” should be modified to “fig. 2”.
  5. Line 263 and line 270: “2N” should be modified to “2n”.
  6. The chromosome at the bottom of Figure 3B is too far away and questionable, please replace the image.
  7. Please improve the clarity of Figure 4.
  8. Is it plausible to use only one gene (plzf) for cell type identification of cells after Percoll isolation?
  9. Line 310-311: “On day 7, five or seven male olivaceus GSCs formed cell aggregates, which were not observed in cultures of OFT or OFO alone.” The formation of cell aggregates does not directly assume that OFT, OFO cells are able to maintain GSCs in cell culture. Is there a separate culture of GSCs as a control? Or do you add more direct evidence that feeder cells can maintain GSCs in vitro culture?
  10. Please add a functional role for the plzf gene in discussion.
  11. I suggest that the gene matching data for 3.2 be displayed in the supplement.
  12. The status and significance of research on gonadal cells as feeder cells should be added to the discussion.
Comments on the Quality of English Language

no

Author Response

Thank you for your valuable review!

Reviewer 2 Report

Comments and Suggestions for Authors

The manuscript "Establishment and Characterization of Testicular and Ovarian Cell Lines from Olive Flounder (Paralichthys olivaceus) for Use as Feeder Cells" by Jo et al. reports the successful establishment of testicular and ovarian cell lines from male and female P. olivaceus, respectively, along with optimized conditions for the enrichment of male P. olivaceus germline stem cells.

I recommend publication of this paper with a few minor issues being addressed.

Minor points:

Point 1: In Figure 3, please include the scale bar.

Point 2: The text in Figures 4 is too small to read comfortably. Please increase the font size for better clarity.

Author Response

Thank you for your valuable review!

Reviewer 3 Report

Comments and Suggestions for Authors

Dear authors;

Thank you for your fluent writing. I thinks it is a basic study and I enjoyed reading it. There is no question for me but below I provide my commentaries.

  1. it was better to compared male and female GSCs and show that OFT and OFO have the same result in both GSCs or not.
  2. Adding a graphical abstract will improve your manuscript.

Author Response

Thank you for your valuable review!
